# Viewing Knowledge Transfer in Multilingual Machine Translation Through a Representational Lens

**David Stap**    **Vlad Niculae**    **Christof Monz**

Language Technology Lab
University of Amsterdam
{d.stap, v.niculae, c.monz}@uva.nl

## Abstract

We argue that translation quality alone is not a sufficient metric for measuring knowledge transfer in multilingual neural machine translation. To support this claim, we introduce Representational Transfer Potential (RTP), which measures representational similarities between languages. We show that RTP can measure both positive and negative transfer (interference), and find that RTP is strongly correlated with changes in translation quality, indicating that transfer *does* occur. Furthermore, we investigate data and language characteristics that are relevant for transfer, and find that multi-parallel overlap is an important yet under-explored feature. Based on this, we develop a novel training scheme, which uses an auxiliary similarity loss that encourages representations to be more invariant across languages by taking advantage of multi-parallel data. We show that our method yields increased translation quality for low- and mid-resource languages across multiple data and model setups.

## 1   Introduction

Multilingual neural machine translation (mNMT) (Ha et al., 2016; Johnson et al., 2017) can support multiple translation directions in a single model, with low-resource languages benefiting most and high-resource languages degrading in quality (Arivazhagan et al., 2019). However, there is a large discrepancy among low-resource languages, with some languages benefiting a lot, while others see relatively little improvement. Conflicting findings have emerged in cross-lingual knowledge transfer research, leaving the underlying causes for this discrepancy unclear. For example, some studies have found that token overlap can be leveraged to increase translation performance (Patil et al., 2022; Wu and Monz, 2023), while others have found that token overlap is unimportant for cross-lingual transfer (K et al., 2020; Conneau et al., 2020).

In the context of transferring knowledge from a parent translation model to a child model, some research has shown that quality improvements are larger when using a closely related parent (Zoph et al., 2016), while others found that unrelated language pairs can work even better (Kocmi and Bojar, 2018). Another finding is that an English-centric model benefits most from positive transfer for directions *into* English, while improvement in the other directions is modest (Arivazhagan et al., 2019).

One of the most striking observations in the literature is that the improvements of many-to-one mNMT can be explained to a large extent by the increased amount of target data (Fan et al., 2021), rather than by cross-lingual knowledge transfer.

Understanding cross-lingual knowledge transfer in the context of mNMT is an under-explored research direction (Hupkes et al., 2023). Despite some existing studies that have examined mNMT representations, none have yet connected these representations to knowledge transfer. For instance, when translating "voiture" in French and "Auto" in German to "car" in English, one would expect that the cross-attention context vectors for French-English and German-English would be similar. However, Johnson et al. (2017) show that clustering occurs on the *sentence level* rather than the *word level*. Even identical sentences in various languages do not occupy the same position in the representation space (Escolano et al., 2022), and encoder representations are dependent on the target language (Kudugunta et al., 2019) instead of source meaning.

In this paper, we investigate the relationship between cross-lingual transfer and cross-attention similarities between languages, which we formalise as Representational Transfer Potential (RTP). This allows us to reason about knowledge transfer in a way translation quality (BLEU) is unable to capture. We investigate cross-attention because it acts as *bottleneck* between the encoder (mostly respon-

sible for representing the source sentence) and the decoder. We find that RTP can be used to quantify positive, as well as negative transfer (also known as interference). Furthermore, we show that these similarities correlate with improvements in translation quality, indicating that there *is* knowledge transfer, and the improved translation quality is *not* only due to the increased data on the target side.

Our approach allows us to identify the dataset and language characteristics that are relevant for transfer, such as multi-parallel overlap between languages. Based on our findings, we propose a method for training a multilingual translation model using an auxiliary similarity loss that exploits multi-parallel data, thereby increasing the degree of language invariance across source representations. Contrary to common perception, a significant amount of multi-parallel data exists within parallel datasets such as WMT, making it more abundant than commonly assumed (Freitag et al., 2020). Our method works by alternately feeding parallel and multi-parallel batches to a model. For multi-parallel batches, we minimize an auxiliary similarity loss that encourages context vectors, resulting from cross-attention, to be similar. Our results show that this approach leads to increased performance for low-resource languages across multiple data and model setups.

## 2 Analyzing Transfer in Many-to-Many Models

In this section, we aim to delve deeper into the understanding of knowledge transfer across languages in mNMT models, moving beyond the commonly used metric of translation quality as a proxy for transfer. By exploring the relationship between transfer and hidden representations in a multilingual model, we aim to gain insight into why certain languages benefit more from multilingual training (as discussed in Section 3). Furthermore, we aim to develop training strategies that can increase representational similarity and thus enhance knowledge transfer (as outlined in Section 4).

### 2.1 Experimental Setup

**Data** To investigate the relationship between transfer and representation in multilingual machine translation, we conduct our experiments on the TED Talks corpus (Qi et al., 2018). The corpus comprises parallel data from 59 languages and is chosen over other large parallel corpora such as

OPUS-100 (Zhang et al., 2020) due to its high translation quality and inclusion of relatively large portions of *explicit* multi-parallel data, which is an important characteristic for our analysis. We train a many-to-many model on all language pairs that contain English in the source or target, resulting in 116 translation directions. To ensure comparable results, we apply joint subword segmentation (Sennrich et al., 2016) and use a vocabulary size of 32K. We also train and evaluate bilingual baselines using the same setup.

Additionally we evaluate on the out-of-domain FLORES-101 evaluation benchmark (Goyal et al., 2021). Out-of-domain data helps to assess robustness and generalization capabilities, and provides a more realistic measure of how well the system can handle diverse and unexpected inputs. This dataset is completely multi-parallel, which is a necessary property for our analysis. It consists of a dev (997 sentences) and devtest (1012 sentences) split, both of which we combine to enhance the robustness of our findings. Sentences are extracted from English Wikipedia, and translated to 101 languages by professional translators.

**Evaluation** We calculate BLEU scores (Papineni et al., 2002) using sacreBLEU (Post, 2018). [1]

**Models** We train many-to-one and many-to-many Transformer base models (Vaswani et al., 2017). Detailed information about the models and training process can be found in Appendix A.1.

**Results** For evaluation on TED we used tokenized BLEU to be comparable with Neubig and Hu (2018) and Aharoni et al. (2019). Table 1 shows that our many-to-one and many-to-many models obtains comparable or better BLEU scores for X→English directions.

### 2.2 (Dis-)advantages of Multilingual Training

Having validated that our model meets strong baselines, we will use the FLORES-101 (Goyal et al., 2021) evaluation dataset for our subsequent analyses. X→En results are summarized in Table 2. [2] In general, low-resource and mid-resource languages benefit (+8.5 and +4.5 BLEU), and high-resource language scores are weakened (−0.7 BLEU) compared to bilingual baselines. Similar to previous findings (Johnson et al., 2017) we find that a many-to-many setup outperforms a many-to-one setup.

---

[1] nrefs:1|case:mixed|eff:no|tok:13a|smooth:exp|version:2.3.1
[2] For full results on TED, see Table 4

| Train dataset size | Be-En 4.5K | Az-En 5.9K | Gl-En 10K | Sk-En 61K | De-En 167K | It-En 203K | He-En 211K | Ar-En 213K | Avg. 109K |
|---|---|---|---|---|---|---|---|---|---|
| Neubig and Hu (2018) (many-to-one) | 18.3 | 11.7 | 29.1 | 28.3 | – | – | – | – | 21.6 |
| Aharoni et al. (2019) (many-to-many) | 21.7 | 12.8 | 30.7 | 29.5 | 33.0 | 35.1 | 33.2 | 28.3 | 28.04 |
| Ours (many-to-one) | 23.8 | 14.3 | 34.9 | 33.4 | 36.3 | 38.5 | 36.5 | 31.3 | 31.1 |
| Ours (many-to-many) | **24.9** | **15.2** | **36.0** | **34.2** | **37.5** | **39.8** | **37.3** | **32.6** | **32.2** |

Table 1: X→En test BLEU (tokenized) on TED Talks corpus for language pairs from Aharoni et al. (2019).

| | low (<10K) 12 languages | mid (10K-150K) 23 languages | high (>150K) 17 languages |
|---|---|---|---|
| bi | 1.2 | 12.3 | **18.6** |
| m2o | $8.8^{*(12/0)}$ | $14.6^{*(20/3)}$ | $15.0^{*(0/17)}$ |
| m2m | $\mathbf{9.7}^{*(12/0)}$ | $\mathbf{16.8}^{*(21/0)}$ | $17.9^{*(0/15)}$ |

Table 2: X→En BLEU on FLORES-101 for bilingual (bi), many-to-one (m2o) and many-to-many (m2m) models. Results are bucketed by number of training examples in TED. $*(n/m)$ denote the fraction of scores in a bucket the are significantly better ($n$) or worse ($m$) to the bilingual baseline, according to bootstrap resampling.

Low-resource BLEU scores have a large standard deviation ($\pm 6.9$), indicating that some languages benefit much more than others.

## 2.3 Representational View on Transfer

To further investigate the differences between multilingual and bilingual models, we will now focus on understanding the underlying mechanics of knowledge transfer. Using translation quality alone as a measure of knowledge transfer is inadequate, as differences in translation quality can have various causes, such as target data distribution (Fan et al., 2021). Therefore, in the following experiments, we aim to gain deeper insight into the mechanisms behind knowledge transfer in multilingual models, focusing on the many-to-many model, which produced the highest translation scores.

When translating two semantically equivalent sentences from different source languages to the same target language, if the context vectors produced by the cross-attention mechanism are (almost) identical for every decoding timestep, the resulting translations will be the same. However, the reverse may not hold true; it is possible for distinct context vectors to produce the same output, and these variations may correspond to specific aspects of the target language. The question of whether source language invariance is a desirable or even necessary trait for an mNMT model remains unresolved.

**Language invariance** Our goal is to determine the degree of language invariance in the encoder representations of our multilingual model, and how this affects translation quality and transfer. Unlike previous studies that have focused on the investigation of hidden encoder and decoder representations (Kudugunta et al., 2019), we concentrate on cross-attention, which connects the encoder and decoder. To investigate the degree of language invariance, we sample semantically equivalent sentence triples $S$ from dataset $\mathcal{D}$, where $S = \{x^1, x^2, y\}$. Here, $x^1$ and $x^2$ are sentences that originate from two different non-English source languages $\ell$ and $\ell'$, while the language of the target sentence $\ell^\tau$ is always English. We then measure the average cosine similarity of the cross-attention vectors of all sentences in $\ell$ and $\ell'$ at different decoding time steps $t$:

$$\text{xsim}_{(\ell,\ell',\ell^\tau)} = \sum_{S \in \mathcal{D}^*} \frac{1}{t} \sum_t \text{c}(\times_t(x^1, y), \times_t(x^2, y)),$$
(1)

where c is the cosine similarity, $\times_t(\cdot, \cdot)$ is the context vector, i.e., the result of encoder-decoder cross-attention at decoding time step $t$, and $\mathcal{D}^*$ is a subset of $\mathcal{D}$ that consists of sentence triples in source languages $\ell$ and $\ell'$, and target language $\ell^\tau$ (English). We use FLORES-101, consisting of 2,009 multi-parallel sentences. As we need multiple source sentences and a single target sentence, our analysis focuses on many-to-one directions. We only consider cross-attention within the final decoder layer in this analysis, and leave extensions to non-English target languages to future work.

The resulting similarity matrix is displayed in Figure 1 for eight languages. An xsim similarity value of 1 indicates that the encoder representations are identical for all decoding time steps, i.e., the representations are language invariant. Conversely, a low similarity suggests that the representations are dissimilar on average, indicating that they are far from being language invariant. From the matrix, we can observe several patterns. High-resource languages tend to have relatively high similarity with other high-resource languages. For instance, the similarity between French and Portuguese,

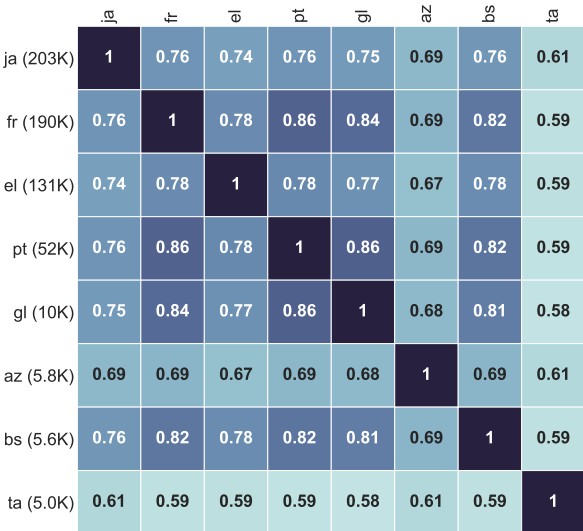

Figure 1: Average cosine similarities between context vectors (see Equation 1) for different source language combinations into English. Train data size is shown between brackets. The higher the similarity, the higher the degree of language invariance.

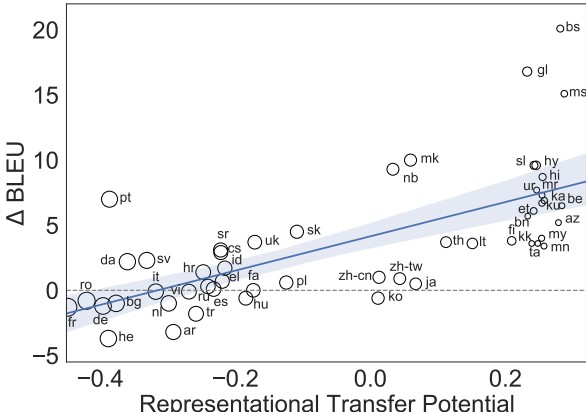

Figure 2: The x-axis represents Representational Transfer Potentials (RTP), which measure the total transfer potential for a language (as detailed in Equation 2), on FLORES-101. The y-axis illustrates the difference in BLEU scores (multilingual BLEU - bilingual BLEU) on FLORES-101. The size of the dots indicates the bilingual BLEU score. The correlation coefficient (Spearman's $\rho$) is .77 and it is statistically significant ($p < 0.001$). The trend illustrates that a higher RTP value is positively associated with changes in translation performance in a multilingual setting.

$\text{xsim}_{(\text{fr, pt, en})}$, is $0.86$, and between Greek and French, $\text{xsim}_{(\text{el, fr, en})}$, is $0.78$. Furthermore, we find that some low-resource languages, such as Galician (gl) and Bosnian (bs), have high similarities with high-resource languages. These languages benefit greatly from multilingual modeling, as evidenced by an increase of 16.8 and 20.1 BLEU points, respectively, compared to their bilingual scores. Other low-resource languages, such as Tamil (ta), do not have high similarities with high-resource languages. These languages do not benefit as much from transfer, as demonstrated by a small increase of only 3.6 BLEU points in the case of Tamil. A full version of the similarity matrix can be found in Appendix B.3.

**Connecting representations to BLEU** We quantify the potential for knowledge transfer into language $\ell \in L$ from other languages $\ell' \in L \setminus \{\ell\}$, by connecting context vector similarity and translation quality. To the best of our knowledge, this is the first approach that quantifies transfer at the representational level. We define the *Representational Transfer Potential* (RTP) as follows:

$$\text{RTP}_{(\ell)} = \sum_{\ell' \in L \setminus \{\ell, \text{en}\}} \frac{\Delta \text{B}(\ell, \ell')}{\max |\Delta \text{B}(\ell, \ell')|} \text{xsim}_{(\ell, \ell', \text{en})}, \quad (2)$$

where $\Delta \text{B}(\ell, \ell')$ is the difference in bilingual BLEU scores between the languages when translat-

ing into English, which can be thought of as an upper bound for the potential transfer between $\ell$ and $\ell'$. $\Delta \text{B}(\ell, \ell')$ is then weighted by the average representational similarity between $\ell$ and $\ell'$ when translating into English, $\text{xsim}_{(\ell, \ell', \text{en})}$ (see Equation 1). RTP thus shows to what extent languages act as donor, i.e., benefiting other languages, or recipient, i.e., benefiting from other languages. Positive transfer can occur when a language $\ell'$ has better translation performance than $\ell$, which increases the weighted $\text{RTP}_{(\ell)}$ score. Negative transfer can occur when language $\ell'$ has worse translation performance than $\ell$, which decreases the score. It is important to note that RTP is *not* a score of a language in isolation, but rather a score of a language dataset in the context of other language datasets. Thus, RTP depends on the languages involved and the available resources in a dataset.

In Figure 2, we plot the resulting RTP scores on the x-axis, and the changes in BLEU scores in a multilingual model versus a bilingual model on the y-axis. We observe a strongly positive and significant correlation ($\rho = .77, p < 0.001$), where a higher RTP score implies increased translation performance, and a lower RTP score implies lower translation performance. Consider Hebrew (he), which has high similarities with lower performing languages, and smaller similarities with better

performing languages. Therefore, RTP can correctly predict that Hebrew *does not* benefit from the multilingual setup, which is evidenced by its negative RTP score ($-.39$) and decreased BLEU score ($-3.7$). On the other hand, Bosnian (bs) has a relatively high RTP score of .28, meaning it is similar to languages with stronger translation quality. Bosnian is the language that benefits most from the multilingual setup ($+20.1$ BLEU). This means that the resulting differences in translation quality are due to knowledge transfer as captured by the RTP score, and not as a side effect of increased target data size. However, it is worth mentioning that this trend is not perfect and can only explain part of the transfer. For instance, Galician and Finnish have similar RTP scores (.23 and .21) but the increase in translation quality for Galician is far greater: 16.8 vs 3.8 for Finnish. The discrepancies in RTP scores warrant further investigation (see next Section).

To ensure the validity and generalizability of our RTP analysis findings beyond a single test dataset (FLORES-101), we incorporate an additional test dataset, NTREX-128 (Federmann et al., 2022). It consists of 1997 multi-parallel sentences in 128 languages. For NTREX-128, we again observe a strongly positive correlation ($\rho = .73, p < 0.001$) between RTP and translation quality, further establishing their relationship. See Appendix B.1 Figure 4 for the corresponding plot. Additionally, the mean absolute RTP deviation per language on FLORES-101 and NTREX-128 is 0.008, and the correlation is extremely robust (($\rho = .99, p < 0.001$)). These results provide further evidence that RTP scores are consistent across different test sets, rather than being an artifact of a specific dataset.

We also perform ablations on RTP and compare to linguistic baselines, which are detailed in Appendix B.1. We conclude that RTP has far better correlation with translation quality compared to ablations and linguistic baselines.

Finally, we show that RTP can be used to pick suitable auxiliary transfer languages. We find that training a language with its top 5 RTP contributors leads to substantially better results of up to 6.8+ BLEU, compared to training with its bottom 5 contributors. More results are in Appendix B.2.

## 3 Analyzing Causes for Transfer

Next, we investigate characteristics that are relevant for transfer. Our objective is to use dataset and linguistic features to predict the representational

similarities $\text{xsim}_{(\ell,\ell',y)}$, as defined in Equation 1.

### 3.1 Data Features and Linguistic Features

**Dataset size:** The difference in training data size for two languages may serve as a predictor for transfer. It is likely that a low-resource language would benefit from a high-resource language. Let $S_\ell$ denote the number of parallel sentences to English for language $\ell$, and $S_{\ell'}$ be defined similarly for language $\ell'$. We then compute the ratio of the smaller value to the larger value as follows:

$$S_{(\ell,\ell')} = \frac{\min(S_\ell, S_{\ell'})}{\max(S_\ell, S_{\ell'})}. \tag{3}$$

Since xsim is symmetric, we design features that are also symmetric, when applicable.

**Vocabulary occupancy:** We calculate the difference in vocabulary occupancy for $\ell$ and $\ell'$. The fraction of the vocabulary that is used by a language captures information about how well the subwords are optimized for that language. Let $V_\ell$ be the set of unique subwords in vocabulary $V$ that are present in the training data $S_\ell$ of language $\ell$. The vocabulary occupancy is then computed as: $|V_\ell|/|V|$. $V_{\ell'}$ is defined similarly. The vocabulary occupancy ratio between $\ell$ and $\ell'$ is defined as:

$$V_{(\ell,\ell')} = \frac{\min(|V_l|/|V|, |V_{\ell'}|/|V|)}{\max(|V_l|/|V|, |V_{\ell'}|/|V|)}. \tag{4}$$

**Source subword overlap:** We measure the similarity between the (subword) vocabularies of language $\ell$ and language $\ell'$. This is calculated by taking the ratio of the number of subwords that are common to both languages ($|V_\ell \cap V_{\ell'}|$) and the total number of unique subwords in both languages ($|V_\ell \cup V_{\ell'}|$) according to the following equation:

$$O_{\text{src}(\ell,\ell')} = \frac{|V_\ell \cap V_{\ell'}|}{|V_\ell \cup V_{\ell'}|}. \tag{5}$$

We also investigated the use of frequency-weighted subwords, which produced similar results.

**Multi-parallel overlap:** We are interested to see how generating identical target sentences (in English) affects transfer. To calculate this, we take the ratio of the number of multi-parallel sentences shared by the languages $\ell$ and $\ell'$, denoted as $S_{\ell'} \cap S_\ell$, to the total number of training sentences in both languages ($S_{\ell'} \cup S_\ell$):

$$S_{\text{shared}(\ell,\ell')} = \frac{|S_{\ell'} \cap S_\ell|}{|S_{\ell'} \cup S_\ell|}. \tag{6}$$

**Target n-gram overlap:** We also measure the similarity between the *generated* target n-grams for the languages. This is similar to the (weighted) source subword overlap but applied to the target side. Let $S_{(\ell,\ell^p)}$ be the set of aligned training sentence pairs of language $\ell$ with pivot language $\ell^p$ (English is taken as pivot here). The (weighted) target subword overlap is then defined as:

$$O_{\text{tgt}(\ell,\ell')} = \sum_i \sum_n n\text{-g}(S^i_{(\ell',\ell^p)}) \cdot n\text{-g}(S^i_{(\ell,\ell^p)}) \cdot n, \tag{7}$$

where $n\text{-g}(\cdot)$ is the n-gram count in a sentence. We have also experimented with higher-order n-grams, and found similar results as unigram, thus we only report the results for unigram.

**Linguistic features:** We adopt five linguistic features, as described in Lin et al. (2019): geographic distance, genetic distance (derived from language descent tree), inventory distance ($k$NN-based phonological inventory vectors, distinct from phonological distance), syntactic distance, and phonological distance.

### 3.2 Experimental Setup

We treat the prediction of the representational similarities $\text{xsim}_{(\ell,\ell',\ell^\tau)}$ (see Equation 1) when translating into target language $\ell^\tau$ (English) between source languages $\ell$ and $\ell'$ as a regression problem. We use the features described in the previous subsection as input variables. To account for variations in feature values across different language pairs, we scale the features between 0 and 1. We consider all 52 source languages. Considering that representational similarities are symmetric, and discarding combinations where $\ell = \ell'$, the resulting number of to be predicted representational similarities is $\frac{(52 \cdot 52) - 52}{2} = 1326$. We use a leave-one-out cross-validation approach, leaving out all similarities for a single language in each round. To evaluate the performance of the model, we use the average (over all language pairs) mean absolute error (MAE) as the evaluation metric. Since different machine learning algorithms have different inductive biases, we train and evaluate three regression models using the scikit-learn library (Pedregosa et al., 2011): linear regression (LR), multilayer perceptron (MLP), and gradient boosting (GB). The detailed hyper-parameter settings used for each model can be found in Appendix A.2.

| | Regressor | LR | MLP | GB |
|---|---|---|---|---|
| | baseline (noise) | 0.061 | 0.061 | 0.061 |
| dataset | dataset size | 0.052 | 0.052 | 0.046 |
| | vocabulary occupancy | 0.041 | 0.041 | 0.035 |
| | multi-parallel overlap | 0.047 | 0.042 | 0.034 |
| | source subword overlap | 0.040 | 0.036 | 0.031 |
| | target subword overlap | 0.050 | 0.046 | 0.042 |
| linguistic | geographic distance | 0.062 | 0.053 | 0.049 |
| | genetic distance | 0.054 | 0.053 | 0.049 |
| | inventory distance | 0.062 | 0.061 | 0.055 |
| | syntactic distance | 0.051 | 0.050 | 0.050 |
| | phonological distance | 0.061 | 0.061 | 0.052 |
| | all data | 0.031 | 0.029 | 0.021 |
| | all linguistic | 0.049 | 0.043 | 0.034 |
| | all data + all linguistic | **0.028** | **0.025** | **0.016** |

Table 3: Mean absolute error (MAE) scores averaged over language pairs for transfer prediction, i.e., predicting $\text{xsim}_{(\ell,\ell',\ell^\tau)}$ (similarity scores between languages $\ell$ and $\ell'$ when translating into English, see Equation 1) using data features and linguistic features (Section 3.1). Best scores per regressor in **bold** and per feature class underlined.

### 3.3 Prediction Results

The results for predicting representational similarities are shown in Table 3. First, combined features lead to better MAE scores than single features. Using all dataset features results in better predictions than using all linguistic features, and combining dataset and linguistic features results in best results for all algorithms. Furthermore, all single features have the potential to improve over a naïve baseline (random input), indicating that they have at least some predictive power.

### 3.4 Feature Importance

We investigate the importance of features to gain a better understanding of their role in transfer.

**Linear regression coefficients:** Weight coefficients are used as a crude measure of feature importance. These coefficients quantify the conditional association between the target xsim and a given feature, while holding other features constant. The sign of the coefficients shows the direction of the association, and the magnitude is an indication of the strength of the association. In Figure 3, we can see that multi-parallel overlap, source subword overlap, and vocabulary occupancy have the largest positive weights among the data features, which implies that these features are positively associated with the target variable and have a strong influence on the prediction. Furthermore, Genetic and Syntactic distance have the highest importance among the linguistic features.

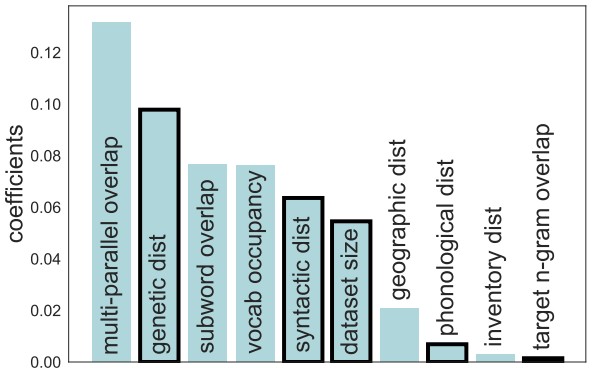

Figure 3: Feature importance for transfer prediction: linear regression sign of coefficients. Absolute values are plotted. Black line indicates negative coefficient (e.g., genetic distance is negative).

**Permutation importance:** To further understand the importance of each feature, we additionally calculate permutation feature importance scores (Breiman, 2001; Fisher et al., 2019). This method evaluates the decrease in model score when a single feature value is randomly shuffled. This model-agnostic procedure breaks the relationship between the feature and the target, thus the drop in the model score is indicative of how much the model depends on the feature. The results using permutation feature importance are consistent with the results obtained using linear regression coefficients. Specifically, we find that multi-parallel overlap is the most important feature for all three regression models. Source subword overlap is also important for MLP and GB, and slightly less for LR. Vocabulary occupancy and dataset size also score relatively high on importance. Genetic distance is consistently the most important linguistic feature among all models. For more details, the permutation feature importance plots can be found in Appendix B.4.

## 4 Optimising for Representational Invariance

Some features that we have shown to be predictive of transfer have been used in previous work. Higher vocab overlap leads to more positive transfer (Chung et al., 2020; Patil et al., 2022; Sun et al., 2022). Temperature sampling addresses dataset size (imbalance) (Arivazhagan et al., 2019). Back-translated data can be used for similar effect (Liao et al., 2021). Grouping languages by their linguistic similarity outperforms English-centric models (Oncevay et al., 2020; Fan et al., 2021). In contrast, there are no such methods for multi-parallel data.

Parallel datasets contain a large number of hidden multi-parallel sentences that remain unused, and resurfacing these improves multilingual translation quality (Freitag and Firat, 2020; Xu et al., 2022). However, these approaches only add multi-parallel data, but do not explicitly exploit multi-parallel properties as part of the learning objective. In contrast, we describe a method that explicitly leverages the characteristics of multi-parallel data.

We introduce an auxiliary similarity loss that encourages context vectors to be more similar when generating the same target token. When sampling a parallel batch, consisting of a source sentence $x$ and the corresponding target sentence $y$, we optimize the cross-entropy loss as usual. When sampling a multi-parallel batch, consisting of meaning equivalent triples $\{x^1, x^2, y\}$ (as defined in Section 2.3), such that $y \neq x^1$ and $x^1 \neq x^2$, we optimize a similarity loss function:

$$\mathcal{L}_{\text{xsim}(x^1,x^2,y)} = \sum_{t=1}^{n} \text{s}(\times_t(x^1, y), \times_t(x^2, y)), \quad (8)$$

where $\text{s}(\cdot, \cdot)$ is a similarity function and $\times_t(\cdot, \cdot)$ is the context vector resulting from the cross-attention at decoding timestep $t$. The goal of minimizing $\mathcal{L}_{\text{xsim}}$ is to encourage representations that are invariant across languages. The final learning objective for multi-parallel batches $(x^1, x^2, y)$ combines minimizing $\mathcal{L}_{\text{xsim}}$ and cross-entropy ($\mathcal{L}_{CE}$):

$$\mathcal{L}_{(x^1,x^2,y)} = \lambda \mathcal{L}_{\text{xsim}(x^1,x^2,y)} + \sum_{i=1}^{2} \mathcal{L}_{\text{CE}(x^i,y)}. \quad (9)$$

### 4.1 Experimental Setup

We follow the setup as described in Section 2.1 and make the following modifications: 1) we sample parallel and multi-parallel batches in a 1:1 ratio, 2) for the multi-parallel batches, we optimize an auxiliary cosine similarity loss and set weight to $\lambda = 1$. To reduce the impact of a small number of dimensions that can dominate similarity metrics, known as rogue dimensions (Timkey and van Schijndel, 2021), we subtract the mean context vector $\bar{c}$ from each context vector in the batch before calculating similarities. If we sample a batch where English is *not* the target, we do not calculate a similarity loss, i.e., $\lambda = 0$. Note, that our method does not require a dataset that is fully multi-parallel. The parallel dataset consists of all X→En and En→X pairs. Its size is 10M pairs. The multi-parallel dataset

|  | FLORES-101 | | | TED | | |
|---|---|---|---|---|---|---|
|  | low (<10K) 12 languages | mid (10K-150K) 23 languages | high (>150K) 17 languages | low (<10K) 12 languages | mid (10K-150K) 23 languages | high (>150K) 17 languages |
| many-to-many | 9.7 | 16.8 | **17.9** | 20.5 | 30.2 | **31.2** |
| + multi-parallel | $9.9^{*(0/0)}$ | $16.9^{*(0/0)}$ | $17.7^{*(0/0)}$ | $20.8^{*(0/0)}$ | $30.1^{*(0/0)}$ | $31.0^{*(0/0)}$ |
| + xsim | $\mathbf{11.5}^{*(12/0)}$ | $\mathbf{17.8}^{*(18/0)}$ | $17.4^{*(0/13)}$ | $\mathbf{21.8}^{*(12/0)}$ | $\mathbf{30.7}^{*(14/1)}$ | $30.4^{*(0/14)}$ |
| many-to-one | 8.8 | 14.6 | **15.0** | 19.9 | 27.9 | **27.2** |
| + multi-parallel | $8.6^{*(0/0)}$ | $14.7^{*(0/0)}$ | $14.9^{*(0/0)}$ | $19.7^{*(0/0)}$ | $27.6^{*(0/0)}$ | $27.0^{*(0/0)}$ |
| + xsim | $\mathbf{10.8}^{*(12/0)}$ | $\mathbf{15.7}^{*(16/2)}$ | $14.5^{*(0/10)}$ | $\mathbf{22.0}^{*(12/0)}$ | $\mathbf{28.8}^{*(14/2)}$ | $26.6^{*(0/11)}$ |

Table 4: X→En BLEU on FLORES-101 and TED test for multilingual many-to-many and many-to-one models, compared to including multi-parallel batches during training (multi-parallel) and additionally adding our auxiliary similarity loss (+ xsim). $*(n/m)$ denote the fraction of scores in a bucket the are significantly better ($n$) or worse ($m$) to the bilingual baseline, according to bootstrap resampling.

consists of all $(x^1, x^2)$ source combinations, with target $y$ fixed to English. The size is 5.9M triples.

Additionally we perform an ablation experiment where we set the similarity loss to 0, to investigate the role of the loss versus the modified data sampling strategy. Note that we cannot ablate the multi-parallel batching, since the similarity loss requires multi-parallel batches.

### 4.2 Results

We include results for our method on both in-domain (TED) and out-of-domain test sets (FLORES-101), for both many-to-many as well as many-to-one models. Table 4 shows BLEU scores and a comparison to the baselines. Adding multi-parallel batches and our similarity loss yields improvements for low- and mid-resource languages, in both many-to-many and many-to-one models. Including multi-parallel batches without applying a similarity loss leads to scores that are not statistically significantly different from the baseline. Furthermore, many-to-many models have the best performance on all aggregated test score buckets. Lowest resource languages benefit most from this approach, with an average BLEU increase of +1.8 and +1.3 (many-to-many). This makes sense, since $\mathcal{L}_{\text{xsim}}$ encourages the representations of these languages to be more similar to other languages, most of which have better performance. Mid-resource languages also benefit from adding $\mathcal{L}_{\text{xsim}}$: +1.0 and +0.5 average increase for FLORES-101 and TED. Higher resource languages suffer from adding the auxiliary loss (−0.5 for FLORES-101, −0.8 for TED). These results demonstrate that lower- and mid-resource languages improve when explicitly optimizing for language invariance using multi-parallel data. Higher-resource languages pay

a small price in performance. This trend holds for in- and out-of-domain test sets, and different types of multilingual models.

## 5 Related Work

**Analyzing mNMT** Investigating representations using Singular Value Canonical Correlation Analysis (SVCCA, Raghu et al., 2017) showed that encoder representations cluster on linguistic similarity, and encoder representations are dependent on the target language (Kudugunta et al., 2019). Additionally, the set of most important attention heads are similar across language pairs, which enables language clustering (Kim et al., 2021). Furthermore, representations of different languages cluster together when they are semantically related (Johnson et al., 2017; Escolano et al., 2019). In particular, visualising cross-attention per decoding time-step shows that meaning equivalent sentences generally cluster together (Johnson et al., 2017).

However, the extent of these phenomena has not been quantified per language. Moreover, these studies have primarily focused on representations in isolation, or its relation with linguistic similarity, with less focus on the role of representations in knowledge transfer. In contrast, we explicitly connect representations to transfer, which allows for a deeper understanding of the impact of transfer on translation quality.

## 6 Conclusion

Previous research has primarily measured knowledge transfer in terms of BLEU scores, leaving open the question of whether improvements in translation quality are due to transfer or other factors such as target data distribution. To address

this gap, we proposed a new measure of knowledge transfer, Representational Transfer Potential (RTP), which measures the representational similarities between languages. We demonstrated that RTP is capable of measuring both positive and negative transfer (interference). A key finding is that RTP is positively correlated with improved translation quality, indicating that the observed improvements in translation quality are a result of knowledge transfer rather than other factors. Additionally, we explored the role of dataset and language characteristics in predicting transfer, and found that multi-parallel overlap is highly predictive for the degree of transfer, yet under-explored in existing literature. We proposed a novel learning objective that explicitly leverages multi-parallel properties, by incorporating an auxiliary similarity loss that encourages representations to be invariant across languages. Our results show that a higher degree of invariance yields substantial improvements in translation quality in low- and mid-resource languages.

## Acknowledgements

This research was funded in part by the Netherlands Organization for Scientific Research (NWO) under project numbers VI.C.192.080 and VI.Veni.212.228. We thank Ali Araabi, Yan Meng, Shaomu Tan and Di Wu for their helpful suggestions and insights.

## Limitations

While our focus is on English-centric many-to-many and many-to-English models, it is important to note that there has been prior work that has explored non-English-centric setups, such as the studies by Fan et al. (2021) and Freitag and Firat (2020). This may present limitations in the generalizability of our results to other multilingual settings. While our analysis already uses 53 languages, we did not measure to what extent our findings hold when using even more languages. Furthermore, our training data size is relatively small which may affect model performance. We use TED instead of the larger OPUS-100 dataset, because TED has higher translation quality and consists of partly multi-parallel data.

## Broader Impact

In general, machine translation poses potential risks such as mistranslation. This risk is higher for low-resource languages. Our method of explicitly aligning representations likely results in less risk for low-resource languages, since the translation quality is improved, and increased risk for high-resource languages.

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

## A   Experimental Setup

### A.1   Translation Models

We use the Transformer Base architecture (6 layers, model dimension 512, hidden dimension 2048 and 8 attention heads) and share all parameters between all language pairs (Ha et al., 2016; Johnson et al., 2017). We use Adam (Kingma and Ba, 2015) ($\beta_1 = 0.9$, $\beta_2 = 0.98$ and $\epsilon = 10^{-9}$) to optimize a label smoothed (Szegedy et al., 2016) (smoothing=0.1) cross entropy loss function. To be able to make use of multilingual data within a single system we use a target-language prefix tag to each source sentence (Johnson et al., 2017; Ha et al., 2016). We tie the weights of the decoder input embeddings and the decoder softmax layer (Press and Wolf, 2017) and apply a 0.2 dropout rate (Srivastava et al., 2014) on the sum of the input- and positional embeddings, on the output of each sublayer, on the output after the ReLU activation in each feedforward sublayer, and to the attention weights. The resulting model has 93M trainable parameters. We use a batch size of 25k tokens. Following Neubig and Hu (2018) and Aharoni et al. (2019), we do not use temperature sampling. Models are implemented in our open-source translation system. All models we train converge in approximately 2 days of training, using 4x NVIDIA TITAN V (12GB) GPUs.

### A.2   Regressors

For MLP and GB, we report the average score over 3 random seeds. We do not report STD as it is negligible.

**MLP** : For the multilayer perceptron, we used hidden layer dimensionality 80 using 3 layers. We use the ReLU activation function, Adam (Kingma and Ba, 2015) ($\beta_1 = 0.9$, $\beta_2 = 0.98$ and $\epsilon = 10^{-9}$).

**GB:**   For the gradient booster, we used squared error, 0.1 learning rate, and 100 estimators.

## B   Additional Results

### B.1   Representational Transfer Potential ablations

Table 5 shows ablations on RTP, and linguistic baselines as described in Lin et al. (2019). We calculate correlation coefficients (Spearman's $\rho$) on the metrics and the difference in BLEU scores (multilingual BLEU - bilingual BLEU) on FLORES-101.

| metric | $\rho$ | $p$ |
|---|---|---|
| RTP | .77 | $< .001$ |
| only $\Delta$ BLEU | .56 | $< .001$ |
| only xsim | 0.28 | $< .05$ |
| genetic distance | $-.11$ | $> .30$ |
| inventory distance | $-.14$ | $> .30$ |
| syntactic distance | .14 | $> .30$ |
| phonological distance | $-.13$ | $> .30$ |
| combined distances | $-.01$ | $> .30$ |

Table 5: RTP ablations and linguistic baselines, calculated on FLORES-101.

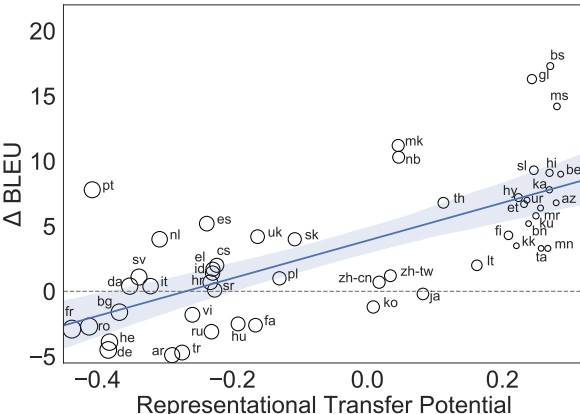

Figure 4: The x-axis represents Representational Transfer Potentials (RTP), which measure the total transfer potential for a language (as detailed in Equation 2), on NTREX-128. The y-axis illustrates the difference in BLEU scores (multilingual BLEU - bilingual BLEU) on NTREX-128. The size of the dots indicates the bilingual BLEU score.

Removing the xsim term in RTP gives $\rho = .56$, and removing $\Delta$ BLEU results in $\rho = 0.77$. The linguistic features do not correlate with BLEU difference. We conclude that RTP has far better correlation with translation quality than linguistic distances and ablations.

Figure 4 shows RTP scores calculated on the NTREX-128 (Federmann et al., 2022) dataset. The trend illustrates that a higher RTP value is positively associated with changes in translation performance in a multilingual setting. The correlation coefficient (Spearman's $\rho$) is 0.73 and it is statistically significant ($p < 0.001$). Figures 4 and 2 (RTP vs delta BLEU on FLORES-101) are highly similar, indicating that RTP generalizes to different test sets.

| | Be RTP$_{top}$ | size | | Be RTP$_{min}$ | size | | Bn RTP$_{top}$ | size | | Bn RTP$_{min}$ | size |
|---|---|---|---|---|---|---|---|---|---|---|---|
| 1 | uk (0.75) | 107K | -1 | bn (0.59) | 3.9K | 1 | hi (0.65) | 178K | -1 | es (0.58) | 195K |
| 2 | ru (0.75) | 206K | -2 | ta (0.6) | 5.1K | 2 | mr (0.65) | 9.3K | -2 | gl (0.58) | 9.9K |
| 3 | bg (0.73) | 172K | -3 | my (0.6) | 19K | 3 | ur (0.62) | 5.7K | -3 | pt (0.58) | 52K |
| 4 | mk (0.72) | 249K | -4 | mr (0.64) | 9.3K | 4 | mn (0.62) | 7.4K | -4 | it (0.59) | 203K |
| 5 | sr (0.72) | 136K | -5 | ur (0.64) | 5.7K | 5 | hy (0.62) | 203K | -5 | nb (0.59) | 16K |

Table 6: RTP top and bottom contributing languages for Belarusian (first two columns) and Bengali (last two columns). Data sizes for the contributing languages into English are shown in columns size. We underline the smallest from the top or bottom, and use this size to subsample the larger one when creating the RTP top and bottom training sets.

## B.2 Represenational Transfer Potential: Top 5 vs Bottom 5

For Belarusian and Bengali, we find the top 5 and bottom 5 contributors to their RTP scores. We then create a training set for both sets, by comparing the top and bottom data sizes and subsampling the largest such that it has the same size as the smallest. This information is presented in Table 6.

We then train a many-to-many system on the resulting datasets, after including Belarusian or Bengali and English. Results can be found in Table 7. We observe large discrepancies in scores for the top 5 and bottom 5 datasets, even though the dataset sizes are identical, for both in-domain (TED) and out-of-domain (FLORES-101) settings. In all cases, the model trained on the top 5 RTP contributors outperforms the one trained on the bottom 5 contributors. The difference is substantial: Be-En on TED with the top RTP contributors scores 16.2 BLEU, whereas the system trained on the bottom contributors results in 9.4 BLEU. These findings show that RTP can be used to identify suitable auxiliary transfer languages.

## B.3 Encoder Representation Similarity

Figure 5 shows cross-attention similarities between all language combinations.

## B.4 Permutation Feature Importances

See Figure 6 for the feature importance box plots.

|  | FLORES-101 | | TED | |
|---|---|---|---|---|
|  | Be-En | Bn-En | Be-En | Bn-En |
| bilingual | 0.5 | 0.4 | 6.1 | 6.8 |
| many-to-many (all) | **7.0** | **6.1** | **24.9** | **19.3** |
| many-to-many (RTP$_{top}$) | **5.4** | **4.1** | **16.2** | **12.9** |
| many-to-many (RTP$_{min}$) | 2.3 | 1.6 | 9.4 | 6.2 |
| $\Delta$ BLEU | +3.1 | +2.5 | +6.8 | +6.7 |

Table 7: Be-En and Bn-En BLEU scores on FLORES-101 and TED. We compare systems trained on bilingual data with many-to-many systems trained on all data (all), the top 5 contributors to RTP (RTP$_{top}$), and the bottom 5 contributors to RTP (RTP$_{min}$).

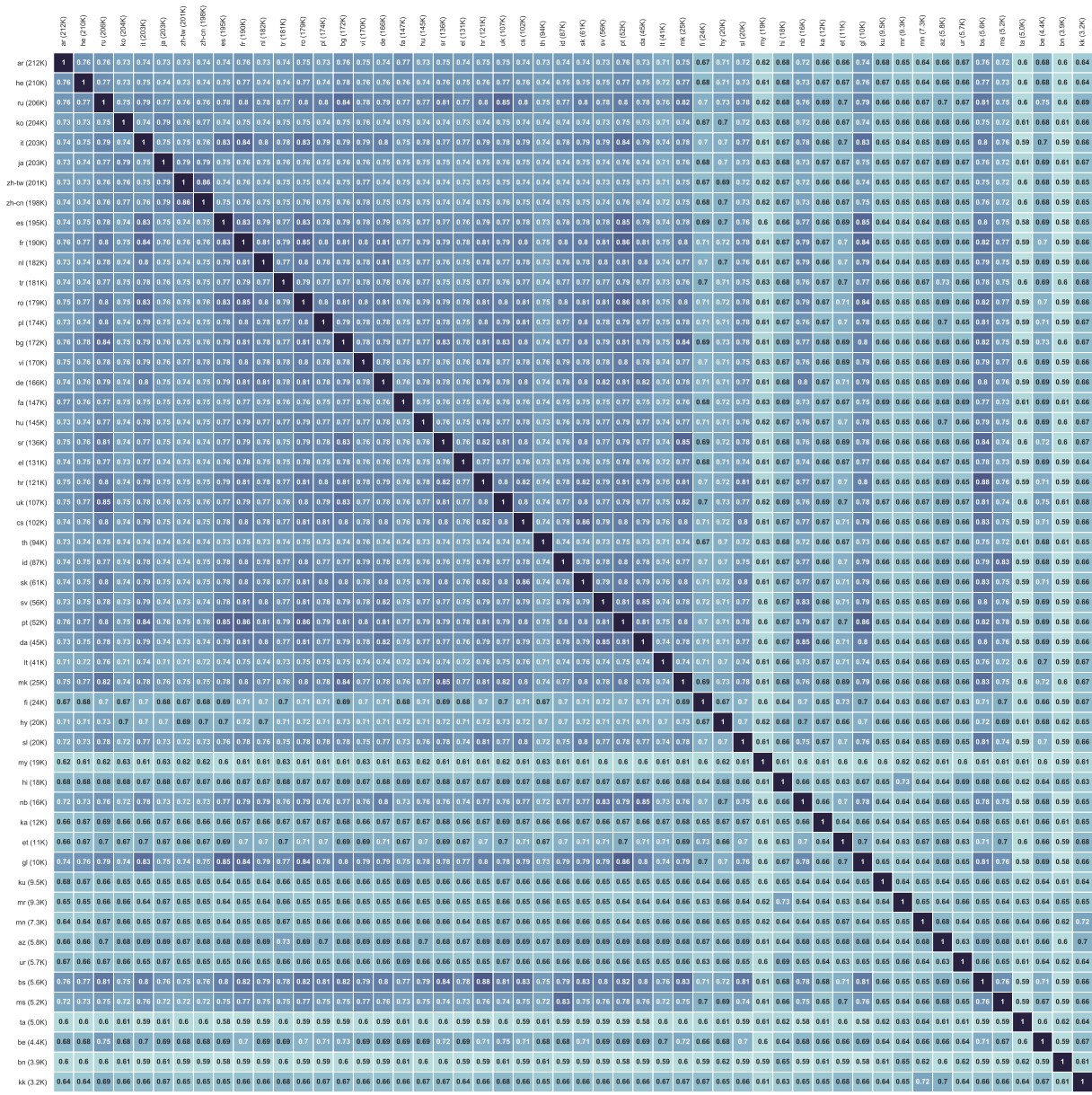

Figure 5: Cross-attention similarities for all language combinations. Training data size into English depicted between brackets. (Zoom for better visibility.)

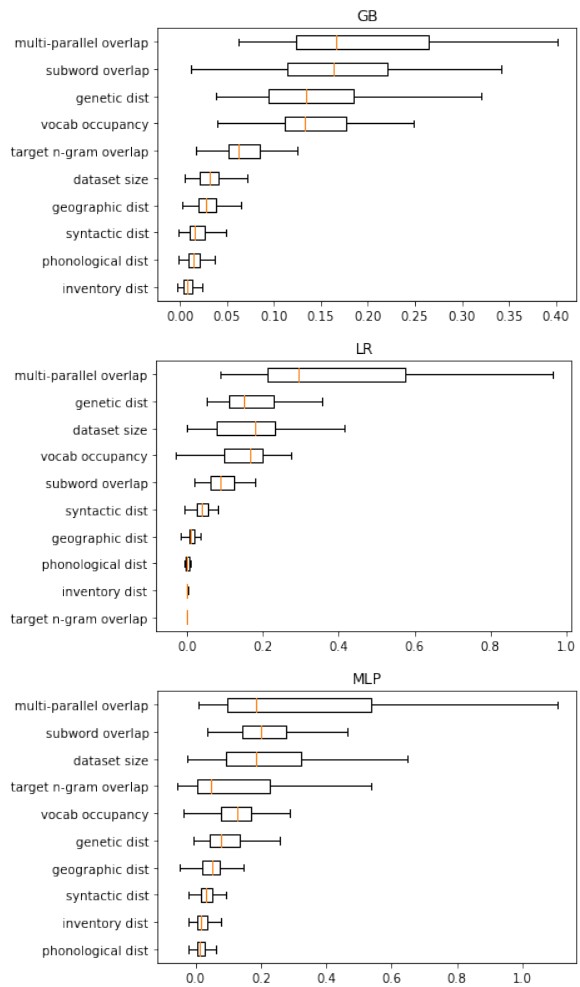

Figure 6: Sorted permutation feature importance scores for LR (top), MLP (middle) and GB (bottom).