# OpenReview forum: "Viewing Knowledge Transfer in Multilingual Machine Translation Through a Representational Lens"
_EMNLP/2023/Conference — EMNLP 2023 Findings_

### Official Review · Reviewer_4uHs · 2023-08-04

**Soundness:** 3

**Excitement:**

3: Ambivalent: It has merits (e.g., it reports state-of-the-art results, the idea is nice), but there are key weaknesses (e.g., it describes incremental work), and it can significantly benefit from another round of revision. However, I won't object to accepting it if my co-reviewers champion it.

**Paper Topic And Main Contributions:**

This work addresses mNMT translation quality under the prism of knowledge transfer.
The question the authors try to solve is whether the improvements in translation quality are due to knowledge transfer or due to other reasons.
To answer this question, this paper proposes a new measure of knowledge transfer called Representational Transfer Potential (RTP) which computes similarities between languages in both positive and negative transfer.
One important finding is that RTP correlates with improved translation quality. Also, the authors explored the role of data sets and language characteristics impact in predicting transfer, showing that multi-parallel overlap is the main indicator for the degree of transfer.
Finally, the authors propose a new learning objective via an auxiliary similarity loss in order to push  representations to be invariant across languages.
The experiments were conducted on two data sets that is: TED Talks and Flores-101.
The results show the effectiveness of the proposed method for low and mid-resource languages. While the results slightly drop for highly-resource languages.






**Questions For The Authors:**

Q1: In Table 1, the baselines are compared to the proposed Many-to-Many model. I was wondering why the authors didn't provide the results of their Many-to-One model?

Q2: This work is an English-centric proposed model as stated in the Limitation Section,  was there any evaluation on pairs of languages not involving English? I was wondering why the target language is always English? Is it because your model is not yet suited for other target languages?

Q3: While there is a little drop in performance for high-resource languages, I was wondering if the authors thought about adapting the  multi-parallel batch and using it only for  low- and mid-resource pairs and see if there is an impact on high resource languages evaluation?

**Reasons To Accept:**

- The paper is clear and well written
- Strong evaluation
- The proposed method shows significant improvements overall (except highly-resource languages)



**Reasons To Reject:**

- No evaluation on other target language than English

**Reproducibility:**

3: Could reproduce the results with some difficulty. The settings of parameters are underspecified or subjectively determined; the training/evaluation data are not widely available.

**Reviewer Confidence:**

3: Pretty sure, but there's a chance I missed something. Although I have a good feel for this area in general, I did not carefully check the paper's details, e.g., the math, experimental design, or novelty.

---

> ### Author Rebuttal · Authors · 2023-08-28
>
> Thank you for your review!
>
> * Q1: In Table 1, the baselines are compared to the proposed Many-to-Many model. I was wondering why the authors didn't provide the results of their Many-to-One model?
>
> We only show the many-to-many (m2m) model and not the many-to-one (m2o) model in Table 1 because the m2m model is stronger (this observation is in line with related work [1]), hence we use that model for subsequent experiments and analyses. For completeness, the m2m vs m2o scores for the languages listed in Table 1 are: Be-En (24.9 vs 23.8), Az-en (15.2 vs 14.3) Gl-En (36.0 vs 34.9) Sk-en (34.2 vs 33.4), De-En (37.5 vs 36.3), It-en (39.8 vs 38.5), He-En (37.3 vs 36.5), Ar-en (32.6 vs 31.3), Average (32.2 vs 31.1). We will include the m2o numbers in Table 1.
>
> * Q2: This work is an English-centric proposed model as stated in the Limitation Section, was there any evaluation on pairs of languages not involving English? I was wondering why the target language is always English? Is it because your model is not yet suited for other target languages?
>
> Multilingual NMT models are often English-centric, and since we investigate this model class in our paper we are limited to English on the source side, or English on the target side. (While these models have 0-shot capabilities, results are generally poor compared to English-centric directions, which is why we excluded them.) Additionally, our analysis uses cross-attention similarities, for which we require sentences from two different source languages, and a sentence from a *fixed* target language. The question we want to answer is: how similar are the 2 source representations when translating into the target? We measure this on the token-level.
>
> * Q3: While there is a little drop in performance for high-resource languages, I was wondering if the authors thought about adapting the multi-parallel batch and using it only for low- and mid-resource pairs and see if there is an impact on high resource languages evaluation?
>
> This is an interesting idea which we also thought about and experimented with, but it led to poor results. We think this is because we do not simply want all representations to be invariant, but we specifically want the “weak” low-resource representations to be similar to the “strong” high-resource representations. This requires multi-parallel batches that mix {low,mid}-resource languages with high-resource languages. In the limit, the {low,mid}-resource sentences are most often combined with high-resource sentences, which draws them closer together. We suspect the high-resource language quality deteriorates a little because their representations are slightly drawn towards the “weak” {low,mid}-resource representations as a result of the xsim loss. We tried to mitigate this using weighting, such that {low,mid}-resource representations are drawn to high-resource but not vice versa, but results did not significantly improve and it complicated our method, so we did not include it in the paper.
>
> Finally, note that we conduct experiments on three (not two) datasets; in addition to TED (for training) and FLORES-101 (evaluation, Wikipedia domain) we also use the NTREX dataset (evaluation, News domain). We used multiple test sets to ensure our method is robust.
>
> [1] Melvin Johnson, Mike Schuster, Quoc V. Le, Maxim Krikun, Yonghui Wu, Zhifeng Chen, Nikhil Thorat, Fernanda Viégas, Martin Wattenberg, Greg Corrado, Macduff Hughes and Jeffrey Dean. Google’s Multilingual Neural Machine Translation System: Enabling Zero-Shot Translation. TACL 2017.

---

### Official Review · Reviewer_kGrk · 2023-08-05

**Soundness:** 2

**Excitement:**

2: Mediocre: This paper makes marginal contributions (vs non-contemporaneous work), so I would rather not see it in the conference.

**Paper Topic And Main Contributions:**

This paper studied knowledge transfer in multilingual translation by analyzing representation similarities between cross-attention outputs. Concretely, it proposes the representational transfer potential (RTP) and analyzes its relation to translation quality. Furthermore, the paper analyzes the origin of transfer and proposes a new auxiliary similarity loss to encourage language-invariant representations and improve translation quality.

**Questions For The Authors:**

- TED seems to be a special case with high-quality multi-parallel data. However, it is not obvious whether there exists sufficient multi-parallel data for large-scale datasets like CCMatrix. Even if they do exist, their quality should also be assessed, as it might be attributed to the artifact of the mining strategy and not true multi-parallel data. Have you inspected this for other datasets?
- In section 2.3, it is stated that "low similarity suggests that the representations are dissimilar on average, indicating that they are far from being language invariant." Based on this, it is not clear why minimizing xsim loss results in invariant representations. Can you provide more detailed explanations?
- Are there any experiment results about ablating xsim loss and multi-parallel batching?

**Reasons To Accept:**

- This paper proposes a novel representation metric to explain multilingual knowledge transfer.
- Analysis for the origin of xsim is provided, and the proposed auxiliary loss results in BLEU improvements for low- and mid-resource languages.

**Reasons To Reject:**

- Two variables in Figure 2 (RTP and BLEU difference) are not fully independent. For example, high-resource languages tend to have low RTP (due to the negative numerator in equation 2) and high BLEU difference (as shown in Table 2). More rigorous justifications should be provided.
- The use of xsim loss is not well motivated. For example, in section 2, only RTP (not xsim) is explicitly connected with BLEU, so the motivation should be stated more concretely.
- Baseline results in Table 4 seem to be the same as the results from section 2. However, the proposed method contains two modifications (different sampling of batches and an additional loss), so ablations for each modification should also be provided to verify the effect of xsim loss.

**Reproducibility:**

4: Could mostly reproduce the results, but there may be some variation because of sample variance or minor variations in their interpretation of the protocol or method.

**Reviewer Confidence:**

4: Quite sure. I tried to check the important points carefully. It's unlikely, though conceivable, that I missed something that should affect my ratings.

---

> ### Author Rebuttal · Authors · 2023-08-28
>
> * Two variables in Figure 2 (RTP and BLEU difference) are not fully independent. For example, high-resource languages tend to have low RTP (due to the negative numerator in equation 2) and high BLEU difference (as shown in Table 2). More rigorous justifications should be provided.
>
> Regarding your comment about RTP and BLEU not being fully independent: this is a good observation, and we agree they are not fully independent. (Note that RTP (Equation 2) uses *bilingual* models and their delta BLEU scores, whereas Figure 2 plots differences with *multilingual* model BLEU scores.) However, RTP ablations in Table 5, Appendix B, indicate that using RTP (i.e., xsim + delta bilingual BLEU) leads to a 0.77 correlation coefficient, whereas removing the xsim part leads to a substantially lower correlation of 0.56. This indicates that the correlation between bilingual and multilingual BLEU can explain only part of the correlation, and including xsim leads to a substantially stronger correlation of 0.77.
>
> * The use of xsim loss is not well motivated. For example, in section 2, only RTP (not xsim) is explicitly connected with BLEU, so the motivation should be stated more concretely.
>
> Regarding your comment about the motivation of xsim: we provide motivation at the beginning of Section 4, see lines 494-513. To summarize, we identified features that are relevant for transfer, and point out related works that have used these features to improve mNMT transfer. We found that multi-parallelism is consistently the best predictor for transfer, and so far no explicit learning objectives have been proposed in the literature that tap into the unique characteristics of this type of data. Our method in Section 4 fills this research gap. (The goal of RTP is to investigate transfer in mNMT models by connecting it to translation quality as measured in BLEU.) We’ll make sure to update the text to make this more clear.
>
> * TED seems to be a special case with high-quality multi-parallel data. However, it is not obvious whether there exists sufficient multi-parallel data for large-scale datasets like CCMatrix. Even if they do exist, their quality should also be assessed, as it might be attributed to the artifact of the mining strategy and not true multi-parallel data. Have you inspected this for other datasets?
>
> Regarding your question about high-quality multi-parallel data: related works [1,2] have shown how to extract multi-parallel sentences in large datasets such as WMT. For example, [1] found that when considering 6 languages in WMT, 123M sentences are available in 2 languages, 6.9M in 3 languages, 5.4M in 4 languages, 0.7M in 5 languages, and 10K in all 6 languages (from Table 3 in [1]). Furthermore, there exist several corpora that are multi-parallel by design, such as UN (5 languages, 11M sentence pairs), Europarl (20+ languages, up to 2M sentence pairs), EMEA (20 languages, 1M sentence pairs). Finally, there are ongoing efforts that investigate multi-parallelism (quantity, quality) for larger scale datasets such as CCMatrix.
>
> * In section 2.3, it is stated that "low similarity suggests that the representations are dissimilar on average, indicating that they are far from being language invariant." Based on this, it is not clear why minimizing xsim loss results in invariant representations. Can you provide more detailed explanations?
>
> Regarding your question about why minimizing xsim loss results in invariant representations: L_{xsim} uses a cosine similarity loss which we point out in the experimental setup, Section 4.1 lines 537-387. When the cosine similarity is 1 (x and y are identical), the loss is 0. We’ll make sure to update L_{xsim} Equation 9 and corresponding text to make this more clear and avoid confusion.
>
> * Are there any experiment results about ablating xsim loss and multi-parallel batching?
>
> Regarding your question about ablating xsim loss and multi-parallel batching: we ran additional experiments *with* multi-parallel batching and *without* xsim loss. To summarize, these results are almost identical to the multilingual baseline. For many-to-many, the scores are as follows (baseline / ablation: only multi-parallel / xsim): low-resource: (9.7/9.9/11.5) ; mid-resource (16.8/16.9/17.8); high-resource (17.9/17.7/17.4). The other ablation (only xsim without multi-parallel batching) is not possible since xsim requires multi-parallel batches; we cannot calculate it otherwise. We’ll make sure to include these results in the updated version of our paper.
>
>
> [1] Markus Freitag and Orhan Firat. Complete multilingual neural machine translation. WMT 2020.
>
> [2] Yulin Xu, Zhen Yang, Fandong Meng, and Jie Zhou. EAG: Extract and Generate Multi-way Aligned Corpus for Complete Multi-lingual Neural Machine Translation. ACL 2022.

---

### Official Review · Reviewer_sd5X · 2023-08-05

**Soundness:** 3

**Excitement:**

3: Ambivalent: It has merits (e.g., it reports state-of-the-art results, the idea is nice), but there are key weaknesses (e.g., it describes incremental work), and it can significantly benefit from another round of revision. However, I won't object to accepting it if my co-reviewers champion it.

**Paper Topic And Main Contributions:**

The authors investigate the correlation between cross-lingual information flow and cross-attention similarity across different languages. Their proposed novel method, called Representational Transfer Potential (RTP), allows for observing knowledge transfer and inferring facts that BLEU scores cannot capture. A significant discovery is that there is a positive correlation between RTP and improved translation quality, suggesting that the enhancements in translation quality are attributable to knowledge transfer rather than other factors. They also propose a novel learning objective that explicitly utilizes multi-parallel properties.

**Questions For The Authors:**

Question A
- If you exclude xsim from equation (2), what impact does it have on the correlation with the difference in BLEU scores?

**Reasons To Accept:**

* They demonstrated the effectiveness of knowledge transfer in low- and middle-resource languages and identified the crucial attribute for knowledge transfer in multilingual translation.

**Reasons To Reject:**

* RTP depends on the performance of bilingual translation models so it is not a robust metric.

**Reproducibility:**

4: Could mostly reproduce the results, but there may be some variation because of sample variance or minor variations in their interpretation of the protocol or method.

**Reviewer Confidence:**

3: Pretty sure, but there's a chance I missed something. Although I have a good feel for this area in general, I did not carefully check the paper's details, e.g., the math, experimental design, or novelty.

---

> ### Author Rebuttal · Authors · 2023-08-28
>
> Thank you for your review!
>
> * RTP depends on the performance of bilingual translation models so it is not a robust metric.
>
> Regarding your comment that RTP depends on the performance of bilingual models and is therefore not robust; this is a valid point and it was also one of our concerns, which is why the paper lists RTP results on two different test sets from different domains: FLORES-101 (Wikipedia) and NTREX (News). Note that the bilingual model scores are, of course, different for FLORES-101 and NTREX. In lines 321-336 we discuss that both FLORES-101 and NTRES show a strongly positive correlation between RTP and translation quality: 0.77 and 0.73 (p<0.001), respectively. Furthermore, the mean absolute RTP deviation per language on these test sets is only 0.008, and the correlation is extremely robust (0.99, p<0.001), which empirically demonstrates the robustness of RTP. Finally, note that we train on TED and choose different domains for the test sets, instead of relying on the TED test data. All RTP results are on out-of-domain FLORES-101 or NTREX, *not* on TED, which further helps to ensure robustness of our findings.
>
> * If you exclude xsim from equation (2), what impact does it have on the correlation with the difference in BLEU scores?
>
> Regarding your question about excluding xsim from Equation 2, this is a good point which we had already included in our submission by doing ablations on RTP (we mention this in lines 337-341 and link to Appendix B.1). To summarize, excluding xsim from RTP results in a moderate correlation of 0.56 (p<0.001), versus a strong correlation of 0.77 (p<0.001) when including xsim. We conclude that RTP has far better correlation with translation quality compared to ablations.

---

### Meta-Review · Area_Chair_KLN5 · 2023-09-17

**Recommendation:** 3

**Metareview:**

This paper shows that cross-attention similarity in a multilingual NMT system is correlated with translation quality, indicating that higher quality is linked to knowledge transfer. It also demonstrates gains on lower-resourced languages from an auxiliary loss function that encourages representational similarity in multi-way parallel data.

Reviewers were mostly convinced by the analysis, finding the proposed cross-attention similarity metric novel, and the demonstrated link between knowledge transfer and quality credible. They also highlighted the positive results from use of the auxiliary representational loss function. They had reservations about potential lack of generality due to reliance on multi-way parallel data, the fact that the similarity metric depends on parallel data, and experiments on into-English directions only. They also raised questions about a number of smaller technical issues.

This seems to be a nice addition to work on multilingual NMT. The demonstration that knowledge transfer is a key determinant of quality, while not surprising, is a useful and solid contribution. Similarly, the loss function, while roughly in line with previous proposals, is novel in exploiting multi-way parallel data, which recent work has shown to be highly beneficial. The authors did an excellent job of addressing reviewers’ questions, and I don’t have any major concerns about the paper’s technical merits.

---

### Decision · Program_Chairs · 2023-10-07

**Decision:**

Accept-Findings

**Comment:**

This paper shows that cross-attention similarity in a multilingual NMT system is correlated with translation quality, indicating that higher quality is linked to knowledge transfer. It also demonstrates gains on lower-resourced languages from an auxiliary loss function that encourages representational similarity in multi-way parallel data.

Reviewers were mostly convinced by the analysis, finding the proposed cross-attention similarity metric novel, and the demonstrated link between knowledge transfer and quality credible. They also highlighted the positive results from use of the auxiliary representational loss function. They had reservations about potential lack of generality due to reliance on multi-way parallel data, the fact that the similarity metric depends on parallel data, and experiments on into-English directions only. They also raised questions about a number of smaller technical issues.

This seems to be a nice addition to work on multilingual NMT. The demonstration that knowledge transfer is a key determinant of quality, while not surprising, is a useful and solid contribution. Similarly, the loss function, while roughly in line with previous proposals, is novel in exploiting multi-way parallel data, which recent work has shown to be highly beneficial. The authors did an excellent job of addressing reviewers’ questions, and I don’t have any major concerns about the paper’s technical merits.